# Clinicopathological Comparison Between *GREB1*- and *ESR1*-Rearranged Uterine Tumors Resembling Ovarian Sex Cord Tumors (UTROSCTs): A Systematic Review

**DOI:** 10.3390/diagnostics15060792

**Published:** 2025-03-20

**Authors:** Livia Maccio, Damiano Arciuolo, Angela Santoro, Antonio Raffone, Diego Raimondo, Susanna Ronchi, Nicoletta D’Alessandris, Giulia Scaglione, Michele Valente, Belen Padial Urtueta, Francesca Addante, Nadine Narducci, Emma Bragantini, Jvan Casarin, Giuseppe Angelico, Stefano La Rosa, Gian Franco Zannoni, Antonio Travaglino

**Affiliations:** 1Surgical Pathology Unit, S. Chiara Hospital, 38122 Trient, Italy; livia.maccio@apss.tn.it (L.M.); emma.bragantini@apss.tn.it (E.B.); 2Unità Operativa Complessa Anatomia Patologica Generale, Dipartimento di Scienze Della Salute Della Donna, del Bambino e di Sanità Pubblica, Fondazione Policlinico Universitario A. Gemelli IRCCS, 00168 Rome, Italy; damianoarciuolo@yahoo.it (D.A.); nicoletta.dalessandris@policlinicogemelli.it (N.D.); giulia.scaglione@policlinicogemelli.it (G.S.); dr.valente.m@gmail.com (M.V.); belen.padialurtueta@policlinicogemelli.it (B.P.U.); francesca.addante@policlinicogemelli.it (F.A.); nadine.narducci@policlinicogemelli.it (N.N.); gianfranco.zannoni@unicatt.it (G.F.Z.); 3Department of Life Sciences and Public Health, Catholic University of Sacred Heart, 00153 Rome, Italy; 4Department of Medical and Surgical Sciences (DIMEC), University of Bologna, 40127 Bologna, Italy; antonio.raffone2@unibo.it; 5Department of Woman, Child and General and Specialized Surgery, University of Campania “Luigi Vanvitelli”, 80138 Naples, Italy; 6Division of Gynecology and Human Reproduction Physiopathology, IRCCS Azienda Ospedaliero-Universitaria di Bologna, 40138 Bologna, Italy; diego.raimondo@aosp.bo.it; 7Pathology Unit, Department of Oncology, ASST Sette Laghi, 21100 Varese, Italy; susanna.ronchi@asst-settelaghi.it (S.R.); stefano.larosa@uninsubria.it (S.L.R.); 8Department of Medicine and Technological Innovation, University of Insubria, 21100 Varese, Italy; jvan.casarin@uninsubria.it; 9Department of Medicine and Surgery, Kore University of Enna, 94100 Enna, Italy; giuangel86@hotmail.it

**Keywords:** UTROSCTs, *GREB1*, *ESR1*, clinicopathological, immunohistochemistry

## Abstract

**Introduction:** Among uterine tumors resembling ovarian sex cord tumors (UTROSCTs), it has been suggested that *GREB1*-rearranged cases are biologically distinct from *ESR1*-rearranged cases and might be considered as a separate entity. **Objectives:** The aim of this systematic review was to assess the difference between *GREB1*- and *ESR1*-rearranged UTROSCTs with regard to several clinico-pathological parameters. **Methods:** Three electronic databases were searched from their inception to February 2025 for all studies assessing the presence of *GREB1* and *ESR1* rearrangements in UTROSCTs. Exclusion criteria comprised overlapping patient data, case reports, and reviews. Statistical analysis was performed to compare clinicopathological variables between *GREB1*- and *ESR1*-rearranged UTROSCTs. Dichotomous variables were compared by using Fisher’s exact test; continuous variables were compared by using Student’s *t*-test. A *p*-value < 0.05 was considered significant. **Results:** Six studies with 88 molecularly classified UTROSCTs were included. A total of 36 cases were *GREB1*-rearranged, and 52 cases were *ESR1*-rearranged. *GREB1*-rearranged UTROSCTs showed a significantly older age (*p* < 0.001), larger tumor size (*p* = 0.002), less common submucosal/polypoid growth (*p* = 0.005), higher mitotic index (*p* = 0.010), more common LVSI (*p* = 0.049), and higher likelihood to undergo hysterectomy (*p* = 0.008) compared to *ESR1*-rearranged cases. No significant differences were detected with regard to margins, cytological atypia, necrosis, retiform pattern, and rhabdoid cells. No significant differences were found in the immunohistochemical expression of any of the assessed markers (wide-spectrum cytokeratins, α-inhibin, calretinin, WT1, CD10, CD56, CD99, smooth muscle actin, desmin, h-caldesmon, Melan-A/MART1, SF1, or Ki67). *GREB1*-rearranged UTROSCTs showed significantly lower disease-free survival compared to *ESR1*-rearranged UTROSTCs (*p* = 0.049). **Conclusions:** In conclusion, *GREB1*-rearranged UTROSCTs occur at an older age, are less likely to display a submucosal/polypoid growth, and exhibit larger size, a higher mitotic index, more common lymphovascular space invasion, and lower disease-free survival compared to *ESR1*-rearranged UTROSCTs. Nonetheless, the similar immunophenotype suggests that they belong to the same tumor family. Further studies are necessary to confirm this point.

## 1. Introduction

Sex cord tumors are a peculiar group of ovarian neoplasms exhibiting granulosa cell or Sertoli cell differentiation. These tumors are characterized by a diverse growth pattern, including diffuse sheets, anastomosing cords and trabeculae, solid and hollow tubules, and micro- and macrofolliculi, retiform structures; accompanying Leydig-like or luteinized stromal cells can be observed. The typical immunophenotypes of sex cord tumors include positivity for inhibin-α, calretinin, SF1, WT1, CD56, and CD99, with variable positivity for several other markers, such as hormone receptors, smooth-muscle actin, desmin, and CD10; Melan-A/MART1 is typically positive in Leydig cells and in steroid cell tumors [1,2]. Uterine tumors showing histomorphological and immunohistochemical features consistent with sex cord tumors were systematically described for the first time in 1976 by Clement and Scully; these neoplasms have been termed “uterine tumors resembling ovarian sex cord tumors” (UTROSCTs). A subset of UTROSCTs (defined “type I” UTROSCTs) was characterized by a sex cord-like component associated with an endometrial stromal component [3]. Molecular studies performed in recent years have shown that type I UTROCTs actually represent endometrial stromal neoplasms, based on the presence of shared *JAZF1* or *PHF1* gene fusions [4,5]. On the other hand, uterine tumors with a pure sex cord pattern (“type II” UTROSCTs, according to Clement and Scully [3]) are still considered as a separate entity and have retained the name “UTROSCTs” [1,4,5]. These neoplasms are regarded as tumors of uncertain malignant potential. In fact, while the vast majority of UTROSCTs show a benign behavior, less than 10% of cases show recurrence and/or metastasis [4]. UTROSCTs have been found to harbor rearrangements of either the *ESR1* gene or *GREB1* gene in most cases, with *NCOA1*-*2*-*3* genes as typical partners of fusion [5]. In 2019, Lee et al. suggested that *GREB1*-rearranged tumors might be different from classical UTROSCTs, as they are more aggressive than *ESR1*-rearranged cases and do not necessarily show a sex cord-like pattern [6]; such difference might have heavy implications in terms of patient management. However, subsequent studies have not clarified whether *GREB1*-rearranged tumors represent a variant of UTROSCTs or a distinct entity [4,7,8,9,10]. Other rearranged genes, such as *CITED2*, *NR4A3*, *GTF2A1*, *SS18*, and *CTNNB1*, were only rarely detected, and their biological significance is currently undefined [6,8,10].

Based on this premise, the aim of this systematic review is to perform a clinicopathological comparison between *GREB1*-rearranged and *ESR1*-rearranged UTROSCTs.

## 2. Materials and Methods

### 2.1. Study Design

This review was designed following the methods of previous studies [11,12]. All review stages (database searching, study selection, data extraction, data analysis) were performed by three independent authors who consulted together at the end of each stage. This review was reported following the protocol of the PRISMA statement [13]. This review is not eligible for inclusion in PROSPERO because it completed data extraction; thus, registration information is not applicable.

### 2.2. Search Strategy and Study Selection

Four electronic databases (PubMed, Scopus, Web of Sciences, Google Scholar) were searched from their inception to March 2024 by using the following combination of keywords: UTROSCT AND GREB1 AND ESR1. An updated research was performed from April 2024 to February 2025. All studies evaluating the presence of *GREB1* and *ESR1* rearrangements in UTROSCT series were included. A priori defined exclusion criteria were as follows: case reports, non-extractable individual data regarding *GREB1* and *ESR1* gene status, selective reporting of a subset of UTROSCT with peculiar features, overlapping patient data, reviews, and guidelines.

### 2.3. Data Extraction

Data were extracted from primary studies, without modification. The PICO [13] results of our review were as follows: P (population) = patients with UTROSCT; I (intervention/risk factor) = presence of *GREB1* rearrangement; C (comparator) = presence of *ESR1* rearrangement; O (outcome) = clinicopathological variables. UTROSCT cases with known molecular status harboring *GREB1* or *ESR1* rearrangement were individually extracted from each study. Cases with unknown molecular status or not harboring *GREB1* or *ESR1* rearrangement were excluded. All available clinicopathological and immunohistochemical variables were extracted and reported in a table. Extracted data included patient age, tumor size, localization, type of treatment, cytological atypia, mitotic index, tumor cell necrosis, lymphovascular space invasion, infiltrative margins, microscopic growth pattern, tumor cell morphology, positivity of immunohistochemical markers, disease-free survival, and recurrence.

### 2.4. Data Analysis

Each clinicopathological variable was compared between *GREB1*- and *ESR1*-rearranged UTROSCTs. For continuous variables (patient age, tumor size, mitotic index, Ki67 labeling index), mean values ± standard error were calculated and compared by using Student’s *t*-test. For discrete variables (localization, type of treatment, cytological atypia, tumor cell necrosis, lymphovascular space invasion, infiltrative margins, microscopic growth pattern, tumor cell morphology, positivity of immunohistochemical markers), Fisher’s exact test was adopted. For disease-free survival, Kaplan–Meier survival analysis with a log-rank test was adopted. In all analyses, a *p*-value < 0.05 was considered significant. Data analysis was performed by using the Statistical Package for Social Science (SPSS) 18.0 package (SPSS Inc., Chicago, IL, USA).

## 3. Results

### 3.1. Study Selection and Characteristics

A total of 95 records were retrieved through database searching. After duplicate removal, 72 records remained. Out of these, 66 were excluded: 27 were reviews, 15 were out of topic, 10 were case reports, 5 were guidelines, 3 did not perform molecular analyses, 2 only assessed *NCOA* genes without partner of fusion, 2 had no extractable individual data, and 2 selectively reported peculiar subsets of UTROSCTs. Five studies [6,7,8,9,10] were included in the review after the first selection; the updated research led to the inclusion of a sixth study [14]. Six studies with 88 molecularly characterized UTROSCTs were finally included in the review. The study selection process is summarized in Figure 1. Characteristics of the included studies are reported in Table 1.

### 3.2. Variables Analyzed

For each molecularly classified UTROSCT, the following clinicopathological data were extracted: age (continuous variable, reported in years), localization (dichotomized as “submucosal” vs. “intramural”), tumor size (continuous variable, reported in cm), primary treatment (dichotomized as “hysterectomy” vs. “no hysterectomy”), tumor margins (dichotomized as “infiltrative” vs. “expansile”), cytological atypia (dichotomized as “at least moderate” vs. “absent/mild”), mitotic index (continuous variable, reported as number of mitotic figures per 10 high-power fields/HPF), coagulative tumor cell necrosis (dichotomized as “present” vs “absent”), lymphovascular space invasion (dichotomized as “present” vs. “absent”), retiform pattern (dichotomized as “present” vs. “absent”), rhabdoid cells (dichotomized as “present” vs. “absent”), disease-free survival (continuous variable, reported in months), and recurrence (dichotomized as “yes” vs. “no”); extracted clinicopathological data are reported in Appendix A. Data regarding immunohistochemical markers expression were dichotomized as “positive” vs. “negative”, with the exception of Ki67, which was reported as the percentage of positive cell nuclei and managed as a continuous variable.

### 3.3. Statistical Analysis

Patients with *GREB1*-rearranged UTROSCTs showed significantly older age than patients with *ESR1*-rearranged UTROSCTs (55.7 ± 11 vs. 39.9 ± 11 years; *p* < 0.001) and were significantly more likely to undergo hysterectomy (92.9% vs. 63.2%; *p* = 0.008).

*GREB1*-rearranged UTROSCTs showed significantly greater tumor diameter than did *ESR1*-rearranged UTROSCTs (7.2 ± 4.3 vs. 3.8 ± 3.3 cm; *p* = 0.002) and were less likely to display a submucosal exophytic growth (23.1% vs. 60%; *p* = 0.005).

Histologically, *GREB1*-rearranged UTROSCTs showed a significantly higher mean mitotic index than did *ESR1*-rearranged UTROSCTs (2.9 ± 3.3 vs. 1.3 ± 1.6 mitotic figures/10HPF; *p* = 0.010), with significantly more common lymphovascular space invasion (21.7% vs. 0%; *p* = 0.049), while no significant differences were detected regarding the presence of at least moderate cytological atypia (57.1% vs. 42.3%; *p* = 0.510), necrosis (3.2% vs. 5%; *p* = 1), infiltrative margins (90% vs. 87%; *p* = 0.699), retiform pattern (19.4% vs. 19.2%; *p* = 1), and rhabdoid cells (13.9% vs. 23.1%; *p* = 0.411).

Immunohistochemically, no significant differences were found between *GREB1*-rearranged and *ESR1*-rearranged UTROSCTs regarding the expression of wide-spectrum cytokeratin (84% vs. 96.7%; *p* = 0.165), α-inhibin (40.7% vs. 61.1%; *p* = 0.132), calretinin (69.2% vs. 89.3%; *p* = 0.095), estrogen receptor (88.9% vs. 96.7%; *p* = 0.850), progesterone receptor (100% vs. 100%; *p* = 1), WT1 (91.3% vs. 95.7%; *p* = 1), CD10 (42.3% vs. 33.3%; *p* = 0.59), CD56 (94.4% vs. 94.1%; *p* = 1), CD99 (85.7% vs. 83.3%; *p* = 1), smooth muscle actin (45.5% vs. 63.6%; *p* = 0.364), desmin (63% vs. 75%; *p* = 1), h-caldesmon (15.8% vs. 15.8%; *p* = 1), Melan-A/MART1 (28.6% vs. 50%; *p* = 0.576), SF1 (50% vs. 100%; *p* = 0.133), and Ki67 (11.8 ± 2.1 vs. 7.1 ± 2.1; *p* = 0.834).

*GREB1*-rearranged UTROSCTs showed significantly shorter disease-free survival than did *ESR1*-rearranged UTROSCTs (95.1 ± 22.3 vs. 218.0 ± 39.2 months; *p* = 0.049) (Figure 2).

The results of the statistical analysis are summarized in Table 2.

## 4. Discussion

This review showed that *GREB1*-rearranged UTROSCTs occur at an older age, are less likely to display a submucosal/polypoid growth, and have larger size, a higher mitotic index, more common lymphovascular space invasion, and lower disease-free survival when compared to *ESR1*-rearranged UTROSCTs, with no significant immunophenotypical differences.

UTROSCT is an uncommon uterine tumor, accounting for <1% of uterine mesenchymal tumors. UTROSCTs typically involve the uterine corpus, presenting in most cases as an intramural mass or less frequently, as a submucosal, subserosal, or endocavitary polypoid mass; a cervical localization is rare but possible [1,14,15,16].

The clinical presentation of UTROSCTs is mostly unspecific. The most common presenting symptom is abnormal uterine bleeding, but it may be asymptomatic or be accompanied by abdominal discomfort or pelvic pain; hormonal disorders have been occasionally described. On ultrasonography and magnetic resonance imaging, a UTROSCT shows no pathognomonic features and is indistinguishable from a uterine leiomyoma, in some cases showing hemorrhage, cystic degeneration, or necrosis. Data regarding serum markers potentially associated with UTROSCT are also scarce and unspecific [17].

A UTROSCT is unencapsulated but may show well-defined margins; sometimes, the incorporation of myometrial smooth muscle bundles may impart a pseudoinfiltrative appearance. Less frequently, true myometrial invasion is present. The architectural pattern of UTROSCTs is highly diverse, including solid nests and sheets, cords, trabeculae, hollow or solid tubules, retiform and/or glomeruloid growth, and pseudoglandular and papillary structures. Tumor cells are usually small, with round to ovoid nuclei and a variable amount of cytoplasm; the neoplastic cells may be frankly epithelioid or have a spindled shape. Cells with small nuclei and abundant clear cytoplasm (luteinized cells) can be observed and are abundant in some cases. Features typically associated with granulosa cell tumors, such as nuclear grooves and Call–Exner-like bodies, are possible but uncommon. Tumor cell nuclei are typically monotonous, with mild nuclear hyperchromasia, no evident nucleoli, and a low mitotic index. High-grade nuclear features with a high mitotic index are less common. Similarly, other malignant features, such as coagulative tumor cell necrosis and lymphovascular space invasion, are uncommon. The intervening stroma vary from scant to profuse and may be fibroblastic, hyalinized, or edematous; the stroma may contain lymphocytes, foamy histiocytes, and/or multinucleated giant cells; cholesterol crystals and hemosiderin deposition are possible. Rare cases show heterologous elements [1,14,15].

The immunophenotype of UTROSCTs is also heterogeneous and appears to be less consistent than that of true ovarian sex cord tumors. Despite being regarded as mesenchymal tumors, UTROSCTs have been shown to express wide spectrum cytokeratins, in most cases, with variable intensity [4,9]. This differs from true ovarian sex cord tumors, which typically show a dot-like or globoid paranuclear positivity [1,15,18]. Hormone receptors (estrogen receptor and progesterone receptor) are positive in the vast majority (70-100%) of UTROSCTs, similar to other mesenchymal and epithelial tumors of the female genital tract [1,6,9,15]. The diagnostic usefulness of estrogen and progesterone receptors is therefore limited; however, they might have a value in the treatment decisions, as hormone receptor-positive tumors might be treated with hormonal therapy [19,20]. Regarding sex cord markers, α-inhibin and calretinin have been the most used types in the diagnosis of sex cord tumors. In particular, calretinin is considered a very sensitive marker of sex cord differentiation, being positive in almost all (97%) cases of ovarian sex cord tumors; on the other hand, α-inhibin is considered more specific, showing positivity in about 70% of cases. In UTROSCTs, more than half of cases are negative for inhibin, which therefore appears to display a lower diagnostic value than that noted in ovarian sex cord tumors. Moreover, about 30% of cases are negative for calretinin, which still remains as one of the most useful immunohistochemical markers in differential diagnosis [2,9,15,21]. Similarly, FOXL2 and SF1, which are relevant markers of sex cord tumors, are negative in about half of UTROSCTs [2,22,23]. The most commonly expressed sex cord markers in UTROSCTs are WT1, CD99, and CD56, which are positive in the vast majority (70–100%) of cases. However, these markers are less specific and are expressed in other uterine tumors such as endometrial stromal neoplasms [2,6,8,24,25]. In addition, the most used immunohistochemical markers in the differential diagnosis of endometrial stromal neoplasms, i.e., CD10, is also positive in a significant percentage (almost half) of UTROSCT cases [1,9,24,25]. Smooth muscle markers have also been found to be positive in UTROSCTs. In detail, the most commonly positive smooth muscle marker is desmin (which is expressed in about three-fourths of UTROSCTs), followed by smooth muscle actin (which is positive in about half of UTROSCTs). On the other hand, h-caldesmon (which is more specific for smooth muscle tumors) is only positive in a minority of UTROSCT cases [1,9,10,15]. A subset of UTROSCTs also expresses Melan-A/MART-1, a melanocytic marker which is positive in Leydig cells and in steroid cell tumors [1,2,15]. The labeling index of the proliferation marker Ki67 is highly variable in UTROSCTs [9].

Given the variability in the localization, growth pattern, tumor cell type, nuclear features, and immunophenotype, the diagnosis of UTROSCTs may be challenging. In fact, the differential diagnosis of UTROSCTs is broad and includes several other epithelial and mesenchymal tumors, including endometrial carcinoma, smooth muscle tumors, endometrial stromal tumors, PEComa, and adenosarcoma [1,14,15].

In most cases, UTROSCTs are easy to differentiate from endometrial carcinoma due to the latter showing striking epithelial features. However, cases of UTROSCTs exhibiting epithelioid cells and a pseudoglandular or papillary growth pattern might mimic endometrioid carcinoma. In addition, UTROSCTs often expresses wide-spectrum cytokeratins, as discussed above. Generally, endometrioid carcinoma originates from the endometrial epithelium and shows a higher degree of nuclear atypia, a higher mitotic index, a striking myometrial infiltration, and metaplastic changes such as squamous and mucinous differentiation [1,9,14,15]. A rare variant of endometrioid carcinoma. i.e., corded and hyalinized endometrioid carcinoma, also shows mesenchymal-like components composed of epithelioid or spindled cells arranged in anastomosing cords and immersed in a hyaline to myxoid stroma, potentially showing some resemblance to uterine mesenchymal tumors; although these cases may represent a diagnostic challenge, they can be correctly diagnosed, as they show an evident endometrioid glandular component, which merges imperceptibly with the corded component and shows squamous/morular metaplasia [26].

Uterine smooth muscle tumors may show morphological overlap with UTROSCT. These tumors are, in most cases, composed of long intersecting or haphazard fascicles of spindled cells with eosinophilic cytoplasm, which allows pathologists to recognize their smooth muscle origin. Nonetheless, uterine smooth muscle tumors may show unusual features such as epithelioid cells, myxoid features, plexiform architecture, and dense collagenous stroma. In addition, as discussed above, immunohistochemical smooth muscle markers are not uncommonly expressed in UTROSCTs, making the differential diagnosis more complex. The presence of prominent thick-walled vessels and the absence of sex cord marker expression in immunohistochemistry are important features of smooth muscle tumors, which can be useful in differential diagnosis. Compared to UTROSCTs, uterine leiomyosarcomas more commonly show marked nuclear atypia and pleomorphism, a high mitotic index, diffusely infiltrating borders, and coagulative tumor cell necrosis. The differential diagnosis between UTROSCTs and smooth muscle tumors of uncertain malignant potential (STUMP), which lack the striking malignant features of leiomyosarcomas, might be more difficult [1,9,15].

Endometrial stromal tumors, especially endometrial stromal nodule and low-grade endometrial stromal sarcomas, may mimic UTROSCTs, as they may show areas of sex cord-like differentiation [15]. In fact, endometrial stromal tumors with sex cord-like areas had previously been considered as a subtype of UTROSCT. As discussed above, Clement and Scully defined these tumors as “type I UTROSCT”, while “type II UTROSCT” was used to refer to tumors with a pure sex cord-like growth [3]. To date, endometrial stromal tumors with sex cord-like areas are considered as a different entity from UTROSCTs and are lumped together as conventional endometrial stromal tumors. Ascertaining the presence of an overt endometrial stromal cell component is therefore necessary for a correct diagnosis. In detail, an endometrial stromal component is composed of monotonous oval to spindle cells, with no or minimal cytologic atypia, vesicular chromatin, scant cytoplasm, and a low mitotic index, with a characteristic arterioliform vasculature, resembling normal endometrial stroma. Glands are usually absent, although a glandular or pseudoglandular growth may occasionally be present. The immunophenotype of endometrial stromal tumors may widely overlap with that of UTROSCTs. In fact, both tumors may express estrogen and progesterone receptors, CD10, WT1, wide-spectrum cytokeratins, smooth muscle markers, and sex cord markers [1,9,14,15,25]. When morphological and immunophenotypical features are insufficient for a definitive diagnosis, molecular analysis appears necessary. In fact, endometrial stromal nodules and low-grade endometrial stromal sarcomas show typical gene translocations, including *JAZF1-SUZ12* t(7;17)(p15;q21) fusion (in 50% of cases), *JAZF1-PHF1* t(6;7)(p21;p15) (in 6% of cases), *EPC1-PHF1* t(10;17)(q22;p13) (in 4% of cases), *MEAF6-PHF1* t(1;6)(p34;p21) (in 3% of cases), and *MBTD1-Cxorf67* t(X;17)(p11.2;q21.33) (in 2% of cases) [15,27,28].

High-grade endometrial stromal sarcomas are a group of tumors characterized by round, ovoidal, or spindled cells, with scant to moderate eosinophilic cytoplasm and monotonous nuclei with uniform nuclear atypia. These tumors commonly express poorly specific sex cord markers such as CD56 and CD99. Compared to UTROSCTs, high-grade endometrial stromal sarcomas typically lack the sex cord-like architectural structures and show more evident malignant features, including permeative (tongue-like) myometrial infiltration and a high mitotic index (usually ≥ 10 mitotic figures/10 HPF). Moreover, high-grade endometrial stromal sarcomas are typically positive for cyclin D1, BCOR, and CD117/c-kit. In challenging cases, molecular analysis will identify the typical alterations in high-grade endometrial stromal sarcomas, including *YWHAE::NUTM2A/B* fusion, BCOR rearrangements, BCOR “ITD” (internal tandem duplications of different lengths involving exon 15), or fusions involving *JAZF1* and/or *PHF1* (in cases of low-grade endometrial stromal sarcomas with high-grade transformation) [1,15,29].

Uterine perivascular epithelioid cell tumor (PEComa) is a rare uterine mesenchymal tumor characterized by the coexpression of smooth muscle and melanocytic markers. Morphologically, PEComas may show expansile or infiltrative margins and are constituted by epithelioid cells with clear to eosinophilic granular cytoplasms arranged in sheets and nests. Spindled cells and a corded growth pattern may be encountered but are less common. Uterine PEComa is regarded as a tumor of uncertain malignant potential; cases showing at least three of five crucial features (diameter > 5 cm, high-grade atypia, mitotic index > 1 mitotic figure/50 HPFs, necrosis, lymphovascular space invasion) are considered malignant. Immunohistochemically, PEComas show positivity for smooth muscle markers, which can be expressed in UTROSCTs, and melanocytic markers; of the latter type, Melan-A/MART1 can be positive in UTROSCTs, and thus has no role in the differential diagnosis. On the other hand, the melanocytic marker HMB45 is typically negative in UTROSCTs and therefore, appears as a crucial diagnostic marker in differentiating between the two entities. On a molecular level, the two main alterations of PEComa are *TSC1/TSC2* alterations and *TFE3* fusion, which are not present in UTROSCT [1,15,30].

Müllerian adenosarcoma is a rare biphasic tumor showing a benign Müllerian epithelial component and a malignant component. The epithelial component type may vary (more commonly expressed as the endometrial-type; it may show mucinous, ciliated, and/or squamous differentiation) and is often arranged in glands; the stromal component is cellular, mitotically active, and shows periglandular stromal condensation (“cuffing”). The classical architecture of adenosarcoma is “leaf-like”, with intraglandular projections of cellular stroma resembling phyllodes tumors of the breast. The biological behavior of adenosarcoma is heavily affected by histological features: cases showing stromal overgrowth, high-grade stromal components, diffuse myometrial infiltration, and/or heterologous elements are significantly more aggressive. When the characteristic features of adenosarcoma are well defined, the diagnosis is straightforward. However, in some cases, the leaf-like growth is not evident, and the stroma may lack the typical cellularity and mitotic activity. Moreover, the stroma may show smooth muscle differentiation and sex cord-like areas, further complicating the diagnosis. Immunohistochemistry for CD10 and Ki67 usually marks the periglandular neoplastic stroma; however, the immunophenotype of adenosarcoma may vary, and there are no immunohistochemical hallmarks. Similarly, no specific molecular alterations have been found in adenosarcoma. Therefore, the differential diagnosis of adenosarcoma mainly lies in the morphological recognition of the two components [1,14,15,31].

Molecularly, UTROSCTs harbor genetic rearrangements involving either the *ESR1* gene or *GREB1* gene, with *NCOA1-2-3* genes being the typical partners of fusion; the molecular assessment of these fusions might be useful in the differential diagnosis of challenging cases [5]. Similar to STUMP and PEComa, UTROSCT can be considered as a tumor of uncertain malignant potential, as it behaves in a benign fashion in most cases, with a minority of cases showing malignant behavior [1,4]. Therefore, defining the criteria to predict the biological behavior of UTROSCT appears of paramount importance for patient management. Boyraz et al. tried to define the histopathological criteria to identify malignant UTROSCT cases. They found that all malignant UTROSCTs showed at least three of five crucial features: tumor diameter > 5 cm, at least moderate cytologic atypia, at least three mitotic figures/10 HPFs, infiltrative margins, and necrosis. However, the authors did not correlate these features with the molecular background of UTROSCT, despite acknowledging that *GREB1*-rearranged cases might possibly represent a separate and more aggressive entity [4]. In fact, in *GREB1*-rearranged tumors, it has been reported that sex cord-like features are less represented than in *ESR1*-rearranged tumors; the term “*GREB1*-rearranged sarcoma” has also been proposed to distinguish these tumors from classical UTROSCT. However, it is still unclear whether they represent a variant of UTROSCT or a distinct entity [4,6,7,8,9,10].

In this systematic review, we performed a clinicopathological comparison between *GREB1*- and *ESR1*-rearranged UTROSCTs in order to improve the knowledge about the relationship between these two entities. To the best of our knowledge, this is the first systematic review performed to address this critical point. We found that patients with *GREB1*-rearranged UTROSCTs exhibited a significantly older age than patients with *ESR1*-rearranged UTROSCTs, with a mean difference of 15.8 years. In fact, the mean age at diagnosis in *GREB1*-rearranged UTROSCTs was 55.7 years, which is similar to that for adult granulosa cell tumors [32] and low-grade endometrial stromal sarcomas [33]. Instead, *ESR1*-rearranged UTROSCT occurred at a mean age of 39.9 years, appearing as a tumor that preferentially affected premenopausal women. However, the mean age at diagnosis still appears to be significantly older than in other ovarian sex cord tumors such as Sertoli–Leydig cell tumors (~25 years) [34] and juvenile granulosa cell tumors (~13 years) [35].

*GREB1*-rearranged UTROSCTs were significantly larger than *ESR1*-rearranged UTROSCTs. Remarkably, the mean diameter of GREB1-rearranged UTROSCTs (7.2 cm) was above the cutoff established by Boyraz et al. for the assessment of malignancy (5 cm) [4]. Moreover, *GREB1*-rearranged cases were less likely to be submucosal/polypoid than were *ESR1*-rearranged cases (23.1% vs 60%). The larger size and the preferentially intramural localization might be at least in part responsible for the higher frequency of hysterectomy in the *GREB1*-rearranged group (92.9%) compared to that in the *ESR1*-rearranged group (63.2%).

The frequency of at least moderate nuclear atypia (defined as “significant” atypia in uterine smooth muscle tumors [36]) did not significantly differ between *GREB1*- and *ESR1*-rearranged UTROSCTs (57.1% and 42.3%, respectively). The mean mitotic index was significantly higher in *GREB1*- than in *ESR1*-rearranged UTROSCTs, despite appearing to be relatively low (2.9 mitoses/10HPF) and not exceeding the cutoff proposed by Boyraz et al. (three mitoses/10HPF) [4]. Tumor cell necrosis was present in a minority of cases, with no significant difference between the two groups (3.2% vs. 5%). The vast majority of tumors in both groups showed infiltrative margins (at least focally), with relatively similar frequency (90% *vs* 87%). Lymphovascular space invasion was present in 21.7% of *GREB1*-rearranged tumors and in no *ESR1*-rearranged tumors, suggesting a higher aggressiveness of the former.

Retiform pattern, i.e., a growth pattern resembling *rete ovarii/rete testis*, is a sex cord-like pattern which can be found Sertoli–Leydig cell tumors, especially in *DICER1*-mutant cases (which tend to be more aggressive than *DICER1*-wild-type cases) [1,37]. Our review found that a retiform pattern was present in about one-fourth of UTROSCT cases, with a similar frequency between *GREB1*- and *ESR1*-rearranged cases. Rhabdoid cells are cells with eosinophilic cytoplasm and eccentric nuclei, which have been reported in malignant UTROSCT cases [4,38]. We found that a minority of UTROSCT cases exhibit rhabdoid cells, with no significant difference between *GREB1*- and *ESR1*-rearranged cases.

With regard to prognosis, *GREB1*-rearranged UTROSCTs showed significantly shorter disease-free survival than did *ESR1*-rearranged UTROSCTs (95.1 vs. 218 months), confirming the higher aggressiveness of the former compared to the latter. This result could not have been affected by the type of treatment, since all recurrent cases had undergone hysterectomy. The higher aggressiveness of *GREB1*-rearranged UTROSCTs might have implications for patient management. For instance, *GREB1*-rearranged UTROSCTs might require adjuvant treatment to prevent recurrence. In fact, the vast majority of these tumors are positive for hormone receptors, and hormone therapy has been reported to be a feasible treatment for UTROSCTs [19]. Chemotherapy and target therapies might be reserved for overtly malignant cases, independent of the underlying gene rearrangement; however, the effectiveness of these therapies is controversial [39]. Whether *GREB1*-rearrangement should be a contraindication to fertility-sparing treatment is unclear. Watrowski et al. reasonably suggested considering *GREB1*-rearrangement as a risk factor for treatment failure but not as an absolute contraindication to conservative management [17].

Immunohistochemically, numerous markers were assessed, including markers of sex cord differentiation (α-inhibin, calretinin, SF1, CD56, CD99, WT1), smooth muscle differentiation (smooth muscle actin, desmin, h-caldesmon), epithelial differentiation (wide-spectrum cytokeratin), endometrial stromal differentiation (CD10), hormone receptors, Leydig cell markers (Melan-A/MART1), and proliferation index (Ki67). None of the tested markers showed a significant difference between *GREB1*-rearranged and *ESR1*-rearranged UTROSCTs, demonstrating a substantial immunophenotypical overlap between the two entities. This finding suggests that *GREB1*- and *ESR1*-rearranged tumors belong to the same UTROSCT spectrum. Interestingly, *GREB1*-rearranged cases showed a slightly decreased expression of several markers, such as α-inhibin, calretinin, cytokeratin, smooth muscle actin, and desmin, with a slightly higher expression of Ki67. Despite being non-significant, these differences may suggest that *GREB1*-rearranged tumors represent a group of less differentiated UTROSCTs. This view would be in agreement with the less consistent presence of sex cord-like morphology and the more aggressive behavior compared to *ESR1*-rearranged cases [4,6]. A recent study (included in our review) found two main clusters and four subclusters of UTROSCTs, based on RNA sequencing. However, while all *GREB1*-rearranged tumors fell into the same main cluster, *ESR1*-rearranged tumors were represented in both main clusters and in all four subclusters, suggesting the lack of a substantial difference between the two entities. Moreover, no significant clinicopathological differences were found among the different clusters [14].

Interestingly, both *GREB1*- and *ESR1*-rearrangement have been described in uterine tumors other than UTROSCTs, namely adenosarcoma and leiomyosarcoma. Considering the wide morphological and immunohistochemical overlap between UTROSCTs and other uterine neoplasms, it cannot be excluded that these tumors were in fact UTROSCTs mimicking other entities, as suggested by Kao et al. [40].

A limitation of our study is the impossibility of reviewing the clinicopathological features of the UTROSCT cases reported by the primary studies. Moreover, since the clinicopathological variables were inconsistently reported among the several studies, we were unable to perform a multivariable analysis to assess the impact of the several variables on the prognosis of UTROSCTs. Further multicentric studies are encouraged to define this point.

## 5. Conclusions

UTROSCTs harboring *GREB1*-rearrangement occur at an older age, are less likely to display a submucosal/polypoid growth, and have larger size, a higher mitotic index, more common lymphovascular space invasion, and a more aggressive behavior compared to *ESR1*-rearranged UTROSCTs. The immunophenotypical overlap between the two entities suggests that *GREB1*-rearranged cases are not a completely distinct entity but rather represent less differentiated UTROSCTs. Further studies are necessary in this regard.

## Figures and Tables

**Figure 1 diagnostics-15-00792-f001:**
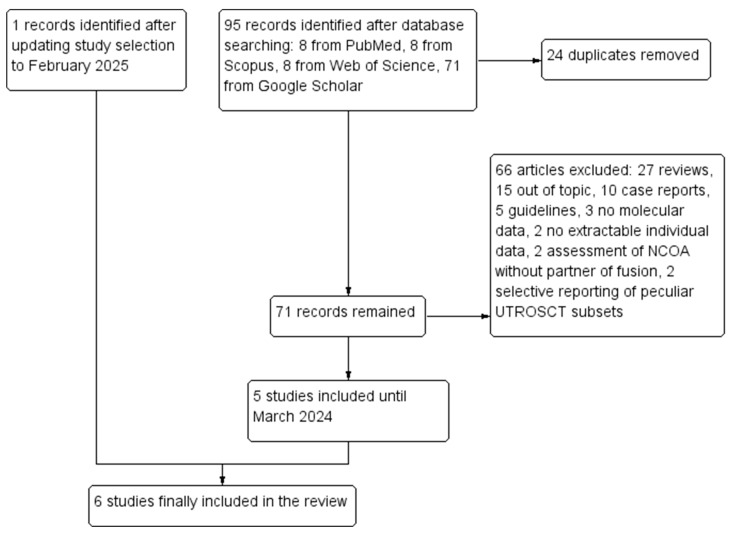
Flow diagram of the study selection process (PRISMA template).

**Figure 2 diagnostics-15-00792-f002:**
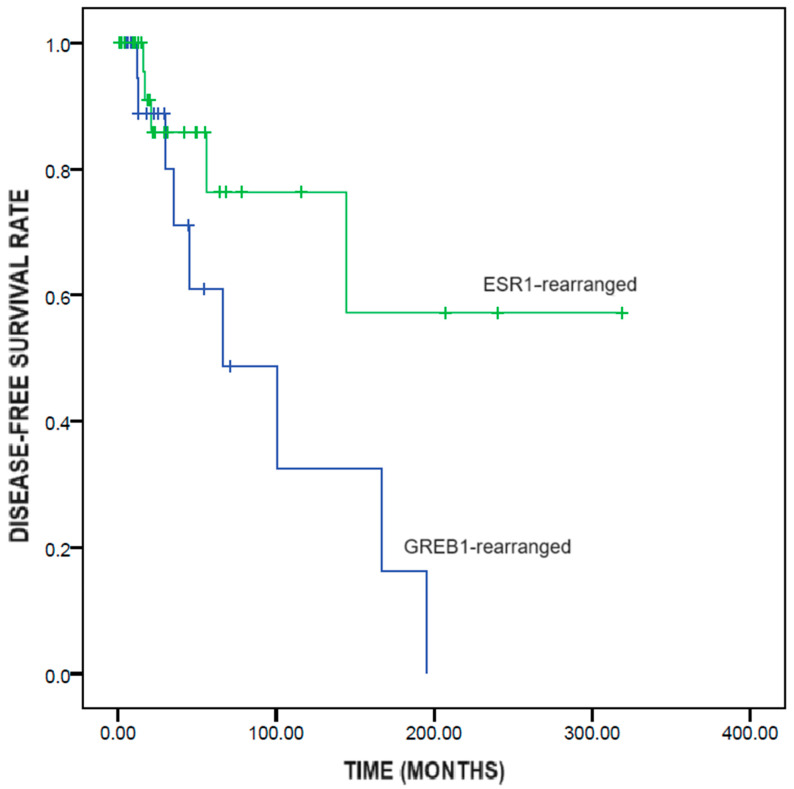
Kaplan–Meier survival analysis of disease-free survival in UTROSCTs according to the genetic rearrangement (*GREB1*-rearranged vs *ESR1*-rearranged).

**Table 1 diagnostics-15-00792-t001:** Characteristics of the included studies.

Study	Year of Publication	Country	Period of Enrollment	Sample Size (No.)	Molecularly Classified Cases	
					*GREB1*-Rearranged	*ESR1*-Rearranged
Lee et al. [6]	2019	Taiwan	2000–2018	8	4	4
Goebel et al. [7]	2020	Usa, Spain, Canada, China	Not reported	27	5	12
Bi et al. [8]	2023	China	2014–2021	23	12	10
Xiong et al. [9]	2023	China	Not reported	19	3	4
Qijun et al. [10]	2024	China	2015–2023	17	4	8
Flidrova et al. [14]	2024	Czech Republic, UK, Italy, Poland	Not reported	35	8	14

**Table 2 diagnostics-15-00792-t002:** Summary of results. * significant *p*-value.

Variable	*GREB1*-Rearranged	*ESR1*-Rearranged	*p*-Value
Age (years)	55.7 ± 11	39.9 ± 11	<0.001 *
Hysterectomy	92.9%	63.2%	0.008 *
Tumor diameter (cm)	7.2 ± 4.3	3.8 ± 3.3	0.002 *
Submucosal localization	23.1%	60%	0.005 *
Mitotic figures/10HPF	2.9 ± 3.3	1.3 ± 1.6	0.010 *
At least moderate atypia	57.1%	42.3%	0.510
Necrosis	3.2%	5%	1
Infiltrative margins	90%	87%	0.699
Lymphovascular space invasion	21.7%	0%	0.049 *
Retiform pattern	19.4%	19.2%	1
Rhabdoid cells	13.9%	23.1%	0.411
Wide spectrum cytokeratin	84%	96.7%	0.165
α-inhibin	40.7%	61.1%	0.132
Calretinin	69.2%	89.3%	0.095
Estrogen receptor	88.9%	96.7%	0.850
Progesterone receptor	100%	100%	1
WT1	91.3%	95.7%	1
CD10	42.3%	33.3%	0.59
CD56	94.4%	94.1%	1
CD99	85.7%	83.3%	1
Smooth muscle actin	45.5%	63.6%	0.364
Desmin	63%	75%	1
h-caldesmon	15.8%	15.8%	1
Melan-A/MART1	28.6%	50%	0.576
SF1	50%	100%	0.133
Ki67 labeling index	11.8 ± 2.1	7.1 ± 2.1	0.834
Disease-free survival (months)	95.1 ± 22.3	218 ± 39.2	0.049 *

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
