# Peer review of "Clinicopathological Comparison Between GREB1- and ESR1-Rearranged Uterine Tumors Resembling Ovarian Sex Cord Tumors (UTROSCTs): A Systematic Review"

_diagnostics, 2025, doi:10.3390/diagnostics15060792_

Round 1
Reviewer 1 Report
Comments and Suggestions for Authors The paper addresses a very important aspect of UTROSCT with far-reaching clinical implications (such as the potential for fertility-sparing therapies). Unfortunately, and unacceptable for a systematic review, the paper excludes almost as many published cases as it includes, which significantly impacts the results, interpretation, and conclusions. Below are my constructive remarks: Major Points: 1. The primary issue is the selection of studies, with only five studies comprising a total of 66 cases. Although the authors state that case reports were excluded (why? - for a systematic review on the molecular aspects of a rare entity, including case reports is both justified and desirable), they also failed to include several case series and studies with up to 35 cases. Her ethe exaples: What is the justification for excluding: a) specific case reports Grither et al., (PMID: 32964092); Devereaux et al., (PMID: 34334688); Yin et al., (PMID: 35081070) b) UTROSCT series with 3 (Bennett et al., PMID: 32675660), 4 (Dickson et al., PMID: 30273195), or 5 patients (Ye et al., PMID: 34348985) c) The omission of the study by Lu et al. (2023, PMID: 36646185), which includes 18 case, many of which demonstrate NCOA1-3 gene rearrangements, is incomprehensible. d) Similarly, the exclusion of the study by Fidorova et al. (PMID: 39265954), which reviews 35 cases, is inacceptable. It cannot be excused by the fact that the initial search ended in March 2024, as the inclusion of this study would have significantly enriched the dataset (adding almost half the number of cases analyzed in the submitted paper). 2. The introduction is too brief and does not adequately reflect the complexity and variety of UTROSCT's clinical presentations. UTROSCT encompasses more than 500 documented cases, as comprehensively reviewed in the literature (PMID: 38276058). In contrast, the authors dive immediately into molecular aspects, neglecting clinical details. For readers unfamiliar with the narrow topic of UTROSCT- which is likely most of them - this makes the relevance of the review unclear! Similarly, it is not evident why the differentiation between GREB1-rearranged and ESR1-rearranged tumors is emphasized while other gene rearrangements (e.g., NCOA1–3, CTNNB1, NR4A3, GTF2A1, CITED2) are considered secondary. What clinical implications do these rearrangements have? These points should be explained and clarified. 3. Here the most importand aspects to be included into the discussion: a) Discuss the implications of GREB1::NCOA-1/3 fusions on the feasibility of fertility-sparing treatments, as proposed by Watrowski et al. (PMID: 38276058) in their seminal work on UTROSCT. b) Address the critical differential diagnostic aspect raised by Kao et al. (PMID: 33099842) specifically the relationship between UTROSCT and other GREB1-rearranged uterine tumors, such as sarcomas and other uterine mesenchymal tumors with similar fusions. Minor Points: Table 1: Due to a formatting error, it is unclear whether Table 1 is complete. This should be reviewed and corrected.Author Response
Comment #1
- A) The paper addresses a very important aspect of UTROSCT with far-reaching clinical implications (such as the potential for fertility-sparing therapies). Unfortunately, and unacceptable for a systematic review, the paper excludes almost as many published cases as it includes, which significantly impacts the results, interpretation, and conclusions. Below are my constructive remarks: Major Points: 1. The primary issue is the selection of studies, with only five studies comprising a total of 66 cases. Although the authors state that case reports were excluded (why? - for a systematic review on the molecular aspects of a rare entity, including case reports is both justified and desirable), they also failed to include several case series and studies with up to 35 cases. Here the exaples: What is the justification for excluding: a) specific case reports Grither et al., (PMID: 32964092); Devereaux et al., (PMID: 34334688); Yin et al., (PMID: 35081070) b) UTROSCT series with 3 (Bennett et al., PMID: 32675660), 4 (Dickson et al., PMID: 30273195), or 5 patients (Ye et al., PMID: 34348985) c) The omission of the study by Lu et al. (2023, PMID: 36646185), which includes 18 case, many of which demonstrate NCOA1-3 gene rearrangements, is incomprehensible. d) Similarly, the exclusion of the study by Fidorova et al. (PMID: 39265954), which reviews 35 cases, is inacceptable. It cannot be excused by the fact that the initial search ended in March 2024, as the inclusion of this study would have significantly enriched the dataset (adding almost half the number of cases analyzed in the submitted paper).
- B) Response: We thank the Reviewer for the Comment. The study by Fidorova et al. was not included because it was published after our study was fully completed. However, as we acknowledge the importance of such publication, we have now updated the analysis and included data from that study, following the Reviewer’s advice.
Regarding the other studies mentioned by the Reviewer, they were intentionally excluded based on our inclusion/exclusion criteria, in order to preserve the integrity of the statistical analysis. In fact, the goal of our review was not to generally describe the clinicopathological features of UTROSCT (there are already other published reviews on this topic, as the Reviewer correctly underlined). Our specific goal (as the title states) was to compare GREB1-rearranged UTROSCT to ESR1-rearranged UTROSCT. This obviously narrows the study selection.
We agree with the Reviewer that case reports may be important in rare entities. However, as our review aimed to compare two populations (GREB1-rearranged UTROSCT and ESR1-rearranged UTROSCT), we believe that it is crucial that every included study encompasses tumors of both populations; this would be impossible for case reports.
Regarding case series, we only included studies that reported clinicopathological data individually or at least separately for ESR1- and GREB1-rearranged tumors. Otherwise, the studies were not suitable for our review as they did not allow comparison between the two groups. This exclusion criterion was specified in our materials and methods section (“individual data regarding GREB1 and ESR1 genes status not extractable”). In the study by Dickson et al., individual clinicopathological data were not reported, so it was impossible to extract data separately for ESR1- and GREB1-rearranged cases. The studies by Ye et al. and Lu et al. reported the presence of NCOA1-3 rearrangement without assessing the partner of fusion. Since the goal of our review was to compare GREB1- and ESR1-rearranged tumors, these studies contained no relevant data to our analysis.
The study by Bennett et al. was excluded because it selectively reported UTROSCT with rhabdoid differentiation and malignant features; this would have obviously introduced a bias in our analysis. This exclusion criterion was specified in our materials and methods section (“selective reporting of subset of UTROSCT with peculiar features”).
Comment #2
- A) The introduction is too brief and does not adequately reflect the complexity and variety of UTROSCT's clinical presentations. UTROSCT encompasses more than 500 documented cases, as comprehensively reviewed in the literature (PMID: 38276058). In contrast, the authors dive immediately into molecular aspects, neglecting clinical details. For readers unfamiliar with the narrow topic of UTROSCT- which is likely most of them - this makes the relevance of the review unclear! Similarly, it is not evident why the differentiation between GREB1-rearranged and ESR1-rearranged tumors is emphasized while other gene rearrangements (e.g., NCOA1–3, CTNNB1, NR4A3, GTF2A1, CITED2) are considered secondary. What clinical implications do these rearrangements have? These points should be explained and clarified.
- B) Response: We thank the Reviewer for the suggestions. We now discussed clinical aspects of UTROSCT in the introduction and, more thoroughly, in the discussion.
Regarding other gene rearrangements, CTNNB1, NR4A3, GTF2A1, SS18 and CITED2 have been described in too few cases to draw any conclusion about their biological significance. By contrast, the vast majority of UTROSCT harbor either GREB1 or ESR1-rearrangement. NCOA1-3 rearrangements are also common; in fact, NCOA1-3 genes are in most cases the partners of fusion of either GREB1 or ESR1. We focused on GREB1- and ESR1-rearranged UTROSCT because there is controversy in the Literature regarding whether they are completely distinct entities or part of the same spectrum. We have now specified these points in the revised manuscript.
Comment #3
- A) Here the most important aspects to be included into the discussion: a) Discuss the implications of GREB1::NCOA-1/3 fusions on the feasibility of fertility-sparing treatments, as proposed by Watrowski et al. (PMID: 38276058) in their seminal work on UTROSCT. b) Address the critical differential diagnostic aspect raised by Kao et al. (PMID: 33099842) specifically the relationship between UTROSCT and other GREB1-rearranged uterine tumors, such as sarcomas and other uterine mesenchymal tumors with similar fusions.
- B) Response: We thank the Reviewer for the suggestions. We have now discussed these points in the revised manuscript.
Comment #4
- A) Minor Points: Table 1: Due to a formatting error, it is unclear whether Table 1 is complete. This should be reviewed and corrected.
- B) Response: We thank the Reviewer for highlighting this issue to us. We have now resolved it in the revised manuscript.
Reviewer 2 Report
Comments and Suggestions for Authors
The article entitled ” Clinicopathological comparison between GREB1- and ESR1-re arranged uterine tumor resembling ovarian sex cord tumor (UTROSCT): a systematic review” is a complet review of the differences between both types of tumors.
It´s a brilliant review in which some differences are shown between this two types of tumors.
- I would like if possible that the authors, could give some clinical information regarding both tumors such as their pattern of appearance, associated symptoms, ultrasound data, diagnostic methods, ... that could help clinicians to identify them and differentiate them from other more frequent pathologies such as uterine fibroids.
- Revise: “Hormone re ceptors (estrogen receptor and progesterone receptor) are positive in the vast majority (70 100%) of UTROSCT, similarly to other mesenchymal and epithelial tumors of the female genital tract [1,6,915].” Please check if it´s 9,15.
- Bibliography: Please revise references 1, 14, 15, 19. Please note that when you reference a book, pages must be included.
Author Response
Comment #1
- A) It´s a brilliant review in which some differences are shown between this two types of tumors.
- B) Response: We thank the Reviewer for the kind comments.
Comment #2
- A) I would like if possible that the authors, could give some clinical information regarding both tumors such as their pattern of appearance, associated symptoms, ultrasound data, diagnostic methods, ... that could help clinicians to identify them and differentiate them from other more frequent pathologies such as uterine fibroids.
- B) Response: We thank the Reviewer for the suggestion. Unfortunately, there are no clinical or imaging data that may help in the diagnosis of UTROSCT, as it appears indistinguishable from uterine leiomyoma. The diagnosis solely relies on pathological examination. We have now discussed these points in the revised manuscript following the Reviewer’s advice.
Comment #3
- A) Revise: “Hormone receptors (estrogen receptor and progesterone receptor) are positive in the vast majority (70 100%) of UTROSCT, similarly to other mesenchymal and epithelial tumors of the female genital tract [1,6,915].” Please check if it´s 9,15.
- B) Response: We thank the Reviewer for highlighting this error to us. We have now corrected it in the revised manuscript.
Comment #4
- A) Bibliography: Please revise references 1, 14, 15, 19. Please note that when you reference a book, pages must be included.
- B) Response: We thank the Reviewer for the suggestion. We have now revised the references and added page numbers.
Round 2
Reviewer 1 Report
Comments and Suggestions for Authors
Thank you for the improvements and the explanations. The mansuript improved greatly.